# LARP7 suppresses P-TEFb activity to inhibit breast cancer progression and metastasis

Xiaodan Ji[1], Huasong Lu[1,2], Qiang Zhou[1], Kunxin Luo[1,3]*

[1]Department of Molecular and Cell Biology, University of California, Berkeley, Berkeley, United States; [2]School of Pharmaceutical Sciences, Xiamen University, Xiamen, China; [3]Life Sciences Division, Lawrence Berkeley National Laboratory, Berkeley, United States

**Abstract** Transcriptional elongation by RNA polymerase (Pol) II is essential for gene expression during cell growth and differentiation. The positive transcription elongation factor b (P-TEFb) stimulates transcriptional elongation by phosphorylating Pol II and antagonizing negative elongation factors. A reservoir of P-TEFb is sequestered in the inactive 7SK snRNP where 7SK snRNA and the La-related protein LARP7 are required for the integrity of this complex. Here, we show that P-TEFb activity is important for the epithelial–mesenchymal transition (EMT) and breast cancer progression. Decreased levels of LARP7 and 7SK snRNA redistribute P-TEFb to the transcriptionally active super elongation complex, resulting in P-TEFb activation and increased transcription of EMT transcription factors, including Slug, FOXC2, ZEB2, and Twist1, to promote breast cancer EMT, invasion, and metastasis. Our data provide the first demonstration that the transcription elongation machinery plays a key role in promoting breast cancer progression by directly controlling the expression of upstream EMT regulators.

*For correspondence: kluo@berkeley.edu

**Competing interests:** The authors declare that no competing interests exist.

**Reviewing editor**: Joaquin M Espinosa, Howard Hughes Medical Institute, University of Colorado, United States

## Introduction

Cancer cells have fundamentally altered gene expression profiles that drive their pathogenic features. In eukaryotes, gene transcription is mainly performed by RNA polymerase (Pol) II and can be controlled at multiple stages including pre-initiation, initiation, elongation, and termination (*Shilatifard et al., 2003*; *Sims et al., 2004*). Transcription elongation has recently been demonstrated to play a critical role in regulating cell growth and differentiation. In Drosophila and human embryonic stem cells, a large number of genes involved in cell growth, renewal, and differentiation are found to be controlled at the elongation stage (*Guenther et al., 2007*; *Muse et al., 2007*; *Zeitlinger et al., 2007*). Moreover, in mammalian cells, the transcription elongation machinery has also been implicated in the regulation of cell proliferation and differentiation (*Zhou and Yik, 2006*; *Romano and Giordano, 2008*).

Shortly after initiation of transcription, Pol II pauses near the transcription start site largely due to the actions of two negative transcription elongation factors NELF and DSIF (*Peterlin and Price, 2006*). The human positive transcription elongation factor b (P-TEFb) reverses this block and stimulates transcriptional elongation by phosphorylating the two negative elongation factors as well as the C-terminal domain (CTD) of the largest subunit of Pol II. These modification events antagonize the actions of the negative elongation factors and also promote co-transcriptional mRNA processing (*Zhou et al., 2012*). P-TEFb is a heterodimer composed of cyclin-dependent kinase 9 (CDK9) and its regulatory partner Cyclin T1 or T2 (CycT1 or T2). The activity of P-TEFb is stringently maintained in a functional equilibrium in cells to accommodate transcriptional demands for different biological activities (*Zhou and Yik, 2006*). Three P-TEFb-containing complexes have been identified, including the inhibitory 7SK snRNP

**eLife digest** To express a gene to make a protein, the gene's DNA must first be transcribed to produce molecules of messenger RNA. The start of the transcription process features two milestones. First, an enzyme called RNA Polymerase II starts the process. Shortly afterwards, however, the process pauses and only starts again when other proteins are recruited. This second step, called transcriptional elongation, is essential for gene expression in cells that are growing and specializing into specific cell types. However, it is unclear how important this second step is for the progression of human cancers, such as breast cancer.

In humans, two proteins join together to form a complex called 'positive transcription elongation factor b' (or P-TEFb for short). This elongation factor encourages the transcriptional elongation step by adding phosphate groups onto RNA Polymerase II and by outcompeting other proteins that act to stop the process. However, some of the P-TEFb proteins in the cell's nucleus are unable to do this because they are held within a complex, which also contains an RNA molecule and some other proteins including one called LARP7. This protein–RNA complex is thought to help to prevent a number of cancers, for example breast cancer or stomach cancer; however the effect of P-TEFb proteins on cancers in humans is not known.

Less LARP7 protein is made in breast cancer cells compared to healthy cells. And when Ji et al. reduced the levels of the LARP7 protein (or the RNA molecule involved in the complex), the P-TEFb proteins were released from the complex and were free to encourage transcriptional elongation. This led to the increased expression of other proteins that switch other genes on or off, including genes that allow breast cancer cells to spread around the body. On the other hand, Ji et al. revealed that freeing the P-TEFb proteins from the complex in the nucleus did not appear to cause new tumors to develop or existing tumors to grow.

Ji et al. suggest that the LARP7 protein normally helps to prevent the spread of breast cancers by keeping the P-TEFb proteins inactive as a part of the protein–RNA complex. One of the next challenges will be to see if drugs that can inhibit the P-TEFb proteins might be useful as new treatments for late stage breast cancer.

complex and two active transcription complexes: the Brd4-P-TEFb complex and the super elongation complex (SEC) (*Zhou et al., 2012*; *Lu et al., 2013b*).

Under normal growth conditions, more than half of nuclear P-TEFb is sequestered in the catalytically inactive 7SK snRNP, which also contains the 7SK snRNA, HEXIM1 (or the homologous HEXIM2), MePCE, and LARP7. This complex represents the major cellular reservoir of inactive P-TEFb (*Zhou and Yik, 2006*; *Zhou et al., 2012*). Within 7SK snRNP, the 7SK snRNA serves as a central scaffold that coordinates key protein–protein interactions and allows HEXIM1/2 to inhibit CDK9 (*Yik et al., 2003*). The La-related protein LARP7 binds to nearly all the nuclear 7SK snRNA via the 3'-UUU-OH sequence and protects it against exonuclease cleavage (*He et al., 2008*; *Krueger et al., 2008*). Stable RNAi-mediated knockdown of LARP7 causes an almost complete depletion of 7SK snRNA, leading to disruption of 7SK snRNP and a shift of P-TEFb toward the active state (*He et al., 2008*). Considering that the 7SK snRNP is the primary source of suppressed P-TEFb, the level of LARP7 directly affects the amount of active P-TEFb, thereby playing a key role in controlling P-TEFb activity.

In response to a number of conditions/agents that globally impact growth and differentiation, P-TEFb is released from the 7SK snRNP and recruited to chromatin templates by the bromodomain protein Brd4, which binds to acetylated histones and the Mediator complex to promote transcription and cell cycle progression (*Yang et al., 2005*; *Mochizuki et al., 2008*; *Yang et al., 2008*). In addition to Brd4, P-TEFb released from 7SK snRNP can also be recruited by a number of gene-specific transcription factors such as the HIV-1 Tat protein and the mixed lineage leukemia (MLL) fusion proteins to form the multi-subunit SEC complex to specifically and efficiently activate their target genes (*He et al., 2010*; *Sobhian et al., 2010*; *Lu et al., 2013a*).

Accumulating data have implicated the involvement of various components of P-TEFb-containing complexes in human cancer. For example, several components of the P-TEFb-containing SEC, such as AFF1, AFF4, ELL1, AF9, and ENL, are all translocation partners of MLL and are important for MLL-based leukemogenesis (*Lin et al., 2010*; *Yokoyama et al., 2010*; *Smith et al., 2011*). In addition, Brd4

has also been considered a promising therapeutic target in acute myeloid leukemia (AML) because of its ability to sustain P-TEFb-dependent *c-Myc* expression (*Blobel et al., 2011*; *Zuber et al., 2011*). Finally, several lines of evidence have implicated the control of P-TEFb by the 7SK snRNP in human breast cancer. First of all, HEXIM1 has been proposed as an inhibitor of breast cell growth since its expression is downregulated by estrogens in breast tumors (*Wittmann et al., 2003*). Moreover, micro-satellite instability (MSI)-induced frameshift mutations in the LARP7 gene have been detected in a significant population of gastric cancer samples, implicating a potential tumor suppressor role of LARP7 in cancers (*Mori et al., 2002*). Consistent with this result, we have previously shown that LARP7 knockdown in the mammary epithelial cell line MCF10A disrupts cell polarity and blocks morpholog-ical differentiation when cultured in the three-dimensional laminin-rich extracellular matrix (3D lrECM) (*He et al., 2008*).

Despite these observations, virtually nothing is known about whether P-TEFb and its associated factors may play a key role during human cancer progression. In this study, we investigated the func-tion of the P-TEFb functional equilibrium in controlling the epithelial–mesenchymal transition (EMT), invasion, and metastasis of human breast cancer. By knocking down LARP7, we released P-TEFb from the 7SK snRNP and stimulated the P-TEFb-dependent transcription of EMT-related genes, resulting in breast cancer EMT and enhanced invasion and metastasis. Our analyses have revealed a strong caus-ative relationship between the invasive phenotypes of human breast cancer and P-TEFb activation by disrupting the 7SK snRNP. Our study has thus provided the first demonstration that the transcription elongation machinery and the P-TEFb network play critical roles in regulating tumor progression, EMT, and metastasis by directly controlling the expression of EMT/metastasis-related genes.

## Results

### LARP7 expression is downregulated in invasive human breast cancer tissues and cells

To investigate whether P-TEFb and its associated factors are involved in human breast cancer progres-sion, we first examined their expression patterns in the publicly accessible Oncomine microarray data-base. Of the known components in the three major P-TEFb-containing complexes, the 7SK snRNP, the Brd4-bound complex, and the SEC, only LARP7 and HEXIM1, two signature components of the 7SK snRNP, showed consistent alteration in human breast cancer tissues. In two independent clinical data sets containing LARP7 information (*Zhao et al., 2004*; *Finak et al., 2008*), LARP7 expression was markedly reduced in breast cancer tissues, especially in the invasive carcinoma, when compared with the matched normal tissues (*Figure 1A*). As downregulation of HEXIM1 in human breast cancer has been reported previously (*Wittmann et al., 2003*), we focused on LARP7 in this study.

We further analyzed the NKI295 breast cancer microarray data set that contains information on the clinical outcomes of patients (*van de Vijver et al., 2002*) to investigate the correlation between LARP7 level and clinical characteristics. Statistical analysis revealed that downregulation of LARP7 correlated with features of advanced cancer progression including estrogen receptor (ER) status, tumor size, and metastasis (*Table 1*). In particular, higher LARP7 levels were observed in older (p=0.003) and ER-positive (p=0.002) patients, and lower LARP7 expression was shown in larger (p=0.012), poorly differentiated (p=0.05) and metastatic (p=0.001) tumors. More importantly, high levels of LARP7 were significantly associated with increased overall survival and recurrence-free survival (*Figure 1B*). Thus, downregula-tion of LARP7 correlates with breast cancer progression, metastasis, and poor prognosis.

We next examined LARP7 protein levels by immunohistochemistry in a human breast cancer tissue array containing 150 duplicated samples of normal human breast tissue, benign tumors, ductal carci-noma in situ (DCIS), and invasive ductal carcinoma of varying pathological grades. In normal mammary tissues, LARP7 was highly expressed in the epithelial cells of the mammary lobuli and terminal ducts (*Figure 1C*). This high expression was also observed in DCIS samples, but significantly attenuated in the invasive ductal carcinoma samples. These data confirm that LARP7 protein levels are also reduced during breast cancer progression, and thus suggest that LARP7 may play a potential tumor suppressor role in breast cancer.

The association between cancer progression and reduced LARP7 expression was also confirmed in a panel of breast cancer cell lines. LARP7 was expressed at relatively high levels in the untransformed mammary epithelial cell lines (MCF10A and EpH4) and noninvasive breast cancer cell lines (MCF7, BT474, T74D, and ZR75B), but was markedly reduced in all four invasive and metastatic cancer lines

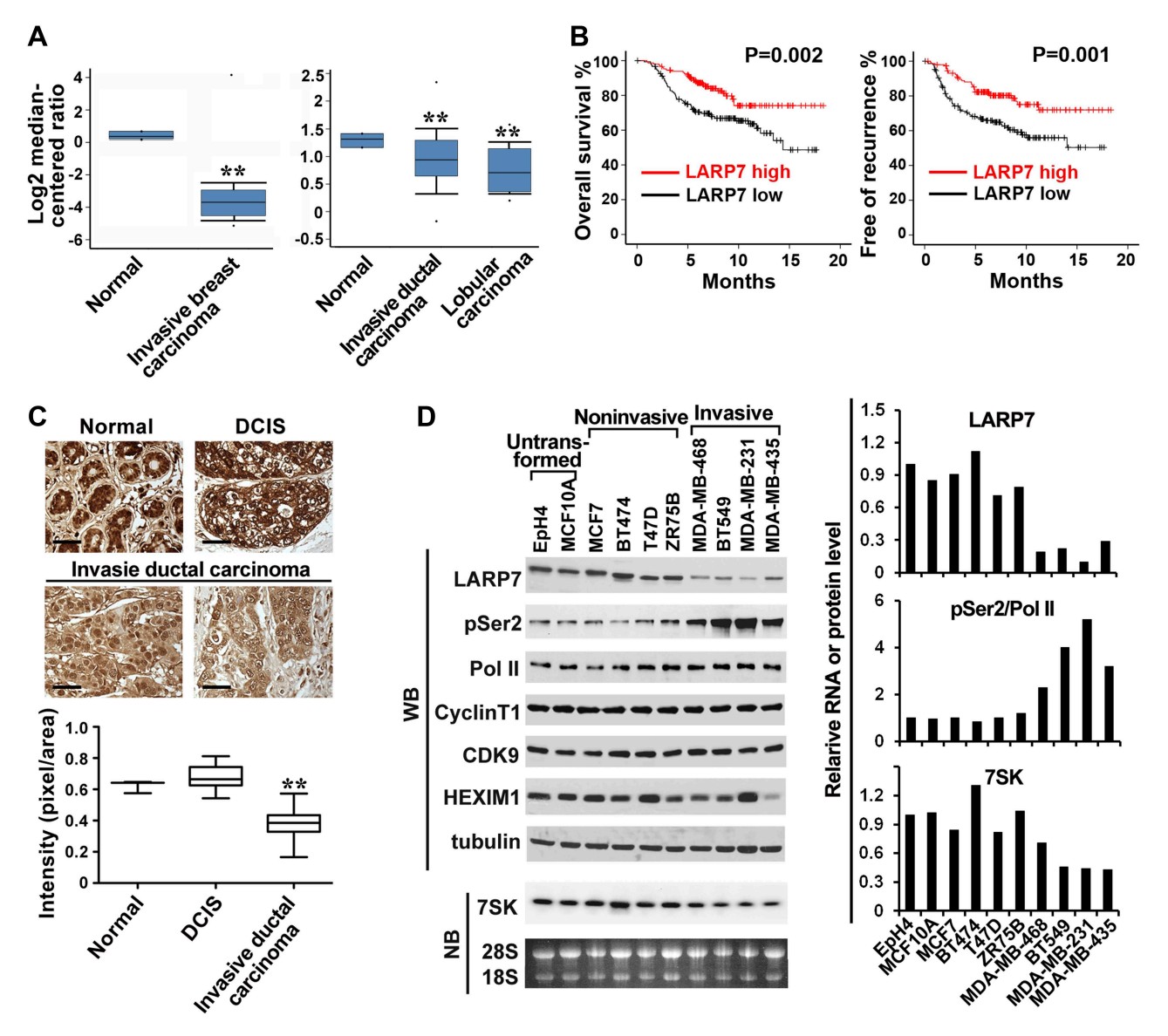

**Figure 1**. LARP7 is significantly downregulated in invasive human breast cancer tissues and cells. (**A**) Box plots show decreased levels of LARP7 in invasive breast carcinoma (left), invasive ductal carcinoma, and lobular carcinoma (right) compared with normal breast tissues in two microarray data sets. **: the p values (p<0.01, compared with normal breast tissues) were determined by the Student's *t* test. (**B**) Kaplan–Meier analysis of overall survival and recurrence-free survival of breast cancer patients stratified by the expression of LARP7. The p values were calculated by the log-rank test. (**C**) Immunohistochemical staining of LARP7 in normal human mammary tissue (n = 6), ductal carcinoma in situ (DCIS) (n = 14), and invasive ductal carcinoma (n = 120). The intensity of LARP7 staining was quantified using ImageJ Plus and shown in the box plot below. Scale bars represent 40 μm. **: the p value (p<0.01, compared with normal breast and DCIS tissues) was determined by the Student's *t* test. (**D**) Western blotting (WB) analysis of the levels of LARP7, phospho-Ser2 (pSer2), total Pol II, CyclinT1, CDK9, and HEXIM1 in various breast cancer cell lines (upper panels) and Northern blotting (NB) analysis of 7SK snRNA levels (lower panels). Tubulin, 28S and 18S RNAs were used as loading controls. Expression of LARP7, pSer2 of Pol II, and 7SK RNA was quantified, normalized to that in EpH4 cells, and shown in the graph to the right.

The following figure supplement is available for figure 1:

**Figure supplement 1**. qRT-PCR analysis of LARP7 mRNA levels in untransformed MCF10A and various breast cancer cell lines.

examined (MDA-MB-468, BT549, MDA-MB-231, and MDA-MB-435; *Figure 1D*). This reduction was partially due to a decrease in the LARP7 mRNA levels as shown by qRT-PCR (*Figure 1—figure supplement 1*). Consistent with the demonstration that LARP7 is necessary for the integrity of the 7SK

**Table 1.** Association between clinical characteristics and LARP7 levels in breast cancer

| Characteristic | Low levels of LARP7 (n = 148) | High levels of LARP7 (n = 147) | p value |
|---|---|---|---|
| Age | | | |
| <45 yr | 95 | 70 | 0.003† |
| ≥45 yr | 53 | 77 | |
| Tumor diameter | | | |
| <20 mm | 45 | 66 | 0.012† |
| ≥20 mm | 103 | 81 | |
| ER status | | | |
| Negative | 46 | 23 | 0.002* |
| Positive | 102 | 124 | |
| No. of positive nodes | | | |
| 0 | 82 | 69 | 0.117 |
| 1–3 | 44 | 61 | |
| ≥4 | 21 | 17 | |
| Differentiation | | | |
| High | 31 | 44 | 0.05† |
| Mediate | 48 | 54 | |
| Poor | 69 | 49 | |
| Metastasis | | | |
| No | 83 | 111 | 0.001* |
| Yes | 65 | 36 | |

*p≤0.01.
†p≤0.05 were determined by chi-square test.

snRNA (*He et al., 2008*), the reduced LARP7 expression in invasive breast cancer cell lines was accompanied by a decrease in 7SK snRNA (*Figure 1D*). Meanwhile, phosphorylation of the Pol II CTD at Ser2 positions (pSer2) strongly increased in the invasive breast cancer cell lines, indicating an increase in the P-TEFb kinase activity. Since the variations in LARP7 expression did not affect the cellular levels of CDK9 and Cyclin T1, this increase in CDK9 kinase activity is likely due to the release of P-TEFb from the inhibitory 7SK snRNP as a result of reduced LARP7 and 7SK snRNA levels. Interestingly, HEXIM1 expression did not correlate with the malignant features of the breast cancer cell lines; it was moderately reduced in three invasive cancer lines and ZR75B non-invasive cancer line, but remained high in the metastatic MDA-MB-231 cells (*Figure 1D*). Taken together, these data suggest a model that LARP7 is downregulated in invasive human breast cancer cells, leading to a decrease in 7SK snRNA and subsequently 7SK snRNP. This results in release of P-TEFb from the inhibitory 7SK snRNP and an increase in the phosphorylation of the P-TEFb substrates. Thus, active P-TEFb appears to play a key role in promoting breast cancer progression.

## LARP7 knockdown (KD) in mammary epithelial cells promotes EMT

To determine the precise role of LARP7 in breast cancer development, we first knocked down the expression of LARP7 in the untransformed MCF10A cells using short hairpin RNAs (shRNAs). Two independent LARP7-targeting sequences (shLARP7-1 and shLARP7-2), when introduced separately into MCF10A cells, markedly decreased the levels of LARP7 protein and 7SK snRNA (*Figure 2A*, upper panel). We have previously reported that these KD cells are partially transformed as evidenced by the disruption of epithelial polarity and morphological differentiation in the 3D lrECM (*He et al., 2008*). During the culture of these cells, we noticed a change in cell morphology from the cobble stone-like shape typical of epithelial cells to a more spindle-like and scattered appearance (*Figure 2A*, lower panel), indicating these cells may be undergoing EMT.

EMT is characterized by a number of functional and molecular changes, including marked increase in cell migration and invasion, actin stress fiber formation, upregulation of mesenchymal markers and

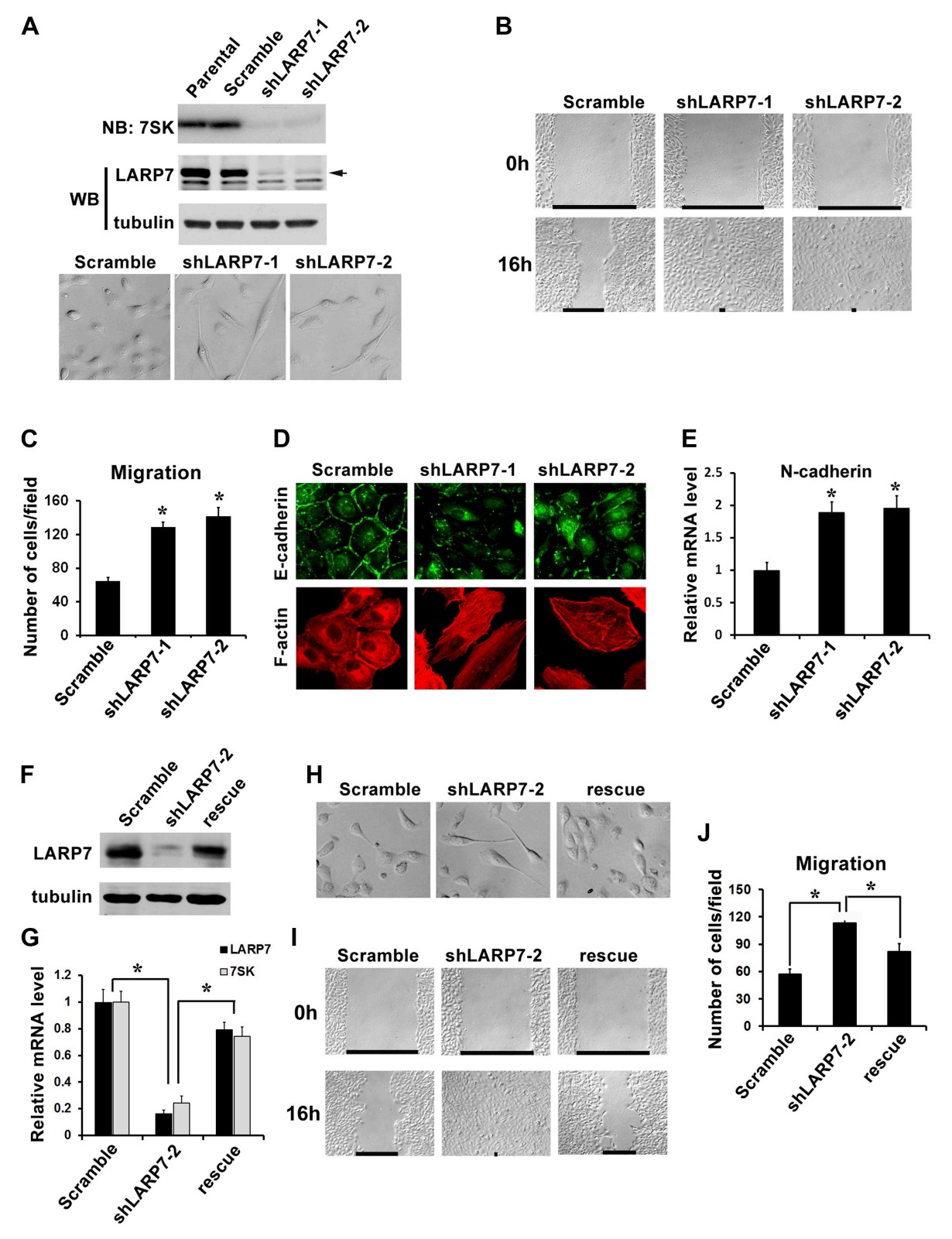

**Figure 2**. Silencing LARP7 induces EMT in MCF10A cells. (**A**) Upper panel, the levels of 7SK snRNA and LARP7 protein in MCF10A parental cells and cells stably expressing scramble control, shLARP7-1, or shLARP7-2 were examined by Northern blotting (NB) and Western blotting (WB), respectively.
*Figure 2. Continued on next page*

*Figure 2. Continued*

Tubulin was used as a loading control. Lower panel, phase-contrast images of control and two shLARP7 pools. (**B**) Wound healing assay. Confluent cell monolayers were wounded, and wound closure was monitored at 0 hr and 16 hr. (**C**) Migration assay. MCF10A control or shLARP7 cells were subjected to a Transwell migration assay. The migrated cells were stained and counted. Data were collected from five fields in three independent experiments. (**D**) Immunofluorescence staining of E-cadherin (green) and actin stress fibers (red). (**E**) qRT-PCR analysis of N-cadherin expression in control and two shLARP7 pools. (**F** and **G**) The levels of LARP7 protein (**F**) and mRNA (**G**) as well as 7SK snRNA level (**G**) in control, shLARP7-2 cells and rescue cells (expressing an shRNA-resistant WT LARP7 cDNA) were examined by Western blotting and qRT-PCR, respectively. (**H**) Phase contrast pictures of control, shLARP7-2, and rescue cells. (**I**) The wound healing assay. (**J**) The Transwell migration assay. For **C**, **E**, **G,** and **J** panels: data are presented as mean ± SD. *: the p values (p<0.05) were determined by the Student's *t* test.

downregulation of epithelial markers (*Thiery et al., 2009*). We therefore measured these characteristics in the MCF10A LARP7 KD cells in order to determine whether the knockdown induces EMT. In the wound healing assay, the KD cells showed markedly faster wound closure than the control cells (*Figure 2B*). Consistently, these cells also displayed significantly accelerated cell migration (*Figure 2C*). In addition, the LARP7 KD cells displayed a marked increase in actin stress fiber formation and a loss of E-cadherin from the adherens junctions (*Figure 2D*). Furthermore, the level of N-cadherin, a mesenchymal marker, was increased nearly twofold upon LARP7 KD (*Figure 2E*). Re-expression of a shRNA-resistant WT LARP7 in the shLARP7-2 cells (*Figure 2F*) effectively restored the loss of 7SK snRNA (*Figure 2G*), restored the epithelial morphology (*Figure 2H*) and rescued the other EMT phenotypes (*Figure 2I,J*), confirming that the enhanced EMT of shLARP7 cells is due to the loss of LARP7 expression. Together, these data indicate that reducing LARP7 promotes EMT in MCF10A cells.

## LARP7 KD in noninvasive breast cancer cell lines promotes malignant progression

In light of the above demonstrations that LARP7 inhibits EMT and that downregulation of LARP7 occurs in invasive, but not noninvasive breast cancer tissues and cell lines, we asked whether downregulation of LARP7 via shRNA could directly promote malignant progression of noninvasive breast cancer cells. To test this, we stably knocked down LARP7 in two noninvasive breast cancer cell lines T47D (*Figure 3A*) and BT474 (*Figure 3—figure supplement 1A*). This resulted in a reduced 7SK snRNA level and caused a significant increase in cell proliferation and anchorage-independent growth in soft agar (*Figure 3B–D*, *Figure 3—figure supplement 1B–D*). The LARP7 KD cells also exhibited dramatically increased EMT as evidenced by accelerated cell migration and invasion (*Figure 3E,F*, *Figure 3—figure supplement 1E*), downregulation of epithelial markers such as E-cadherin, DSP, and KRT19 and upregulation of mesenchymal markers Vimentin, N-cadherin, and matrix metalloproteinases (MMPs) (*Figure 3H*, *Figure 3—figure supplement 1F*). Since EMT has recently been linked to the expansion of cancer stem cells (CSCs) (*Mani et al., 2008*; *Bessede et al., 2013*), we also examined whether the LARP7 KD cells displayed an increased CSC feature in the two-round mammosphere formation assay. Indeed, the KD cells generated bigger and larger number of mammospheres (*Figure 3G*) and displayed elevated levels of various stem cell markers including Oct4, Sox2, and ALDH1 (*Figure 3H*).

Finally, to determine whether LARP7 KD also results in an increase in breast cancer metastasis in vivo, the T47D control or shLARP7-2 cells were injected intravenously into the nude mice, and lung metastasis was examined 12 weeks later. Histological analyses revealed a significant increase in the number of metastatic lesions produced by shLARP7-2 cells when compared with that produced by the control cells (*Figure 3I*). Taken together, our results suggest that knockdown of LARP7 enhances breast cancer EMT and CSC expansion in vitro and metastasis in vivo.

## LARP7 suppresses cell migration through inhibiting P-TEFb

Because LARP7 is a critical component of the inhibitory 7SK snRNP, we hypothesize that its ability to suppress tumor progression is due to the sequestration of P-TEFb in 7SK snRNP. To test this hypothesis, we first examined the kinase activity of P-TEFb upon LARP7 KD in an in vitro kinase assay using recombinant GST-CTD as a substrate. As predicted, the ability of endogenous CDK9 to phosphorylate the CTD was dramatically higher in the KD cells than in the control cells (*Figure 4A*).

If the increased P-TEFb activity upon LARP7 KD is responsible for the EMT and enhanced transformation of breast cancer cells, inhibition of P-TEFb is expected to reverse the process. To test this, we employed flavopiridol, a known P-TEFb inhibitor, which effectively inhibits phosphorylation of CTD

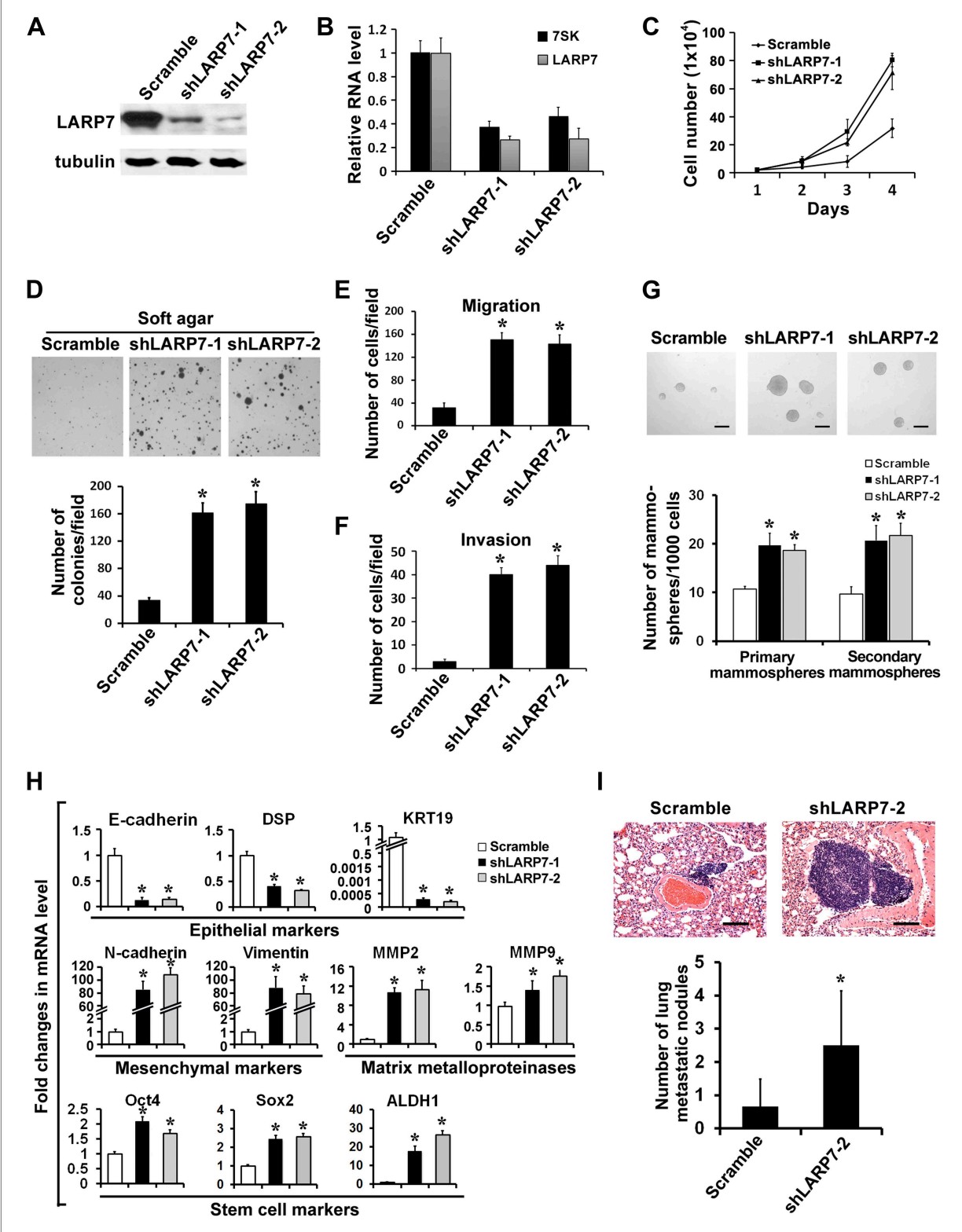

**Figure 3**. Silencing LARP7 promotes malignant progression of T47D breast cancer cells. (**A**) Western blotting shows effective knockdown of LARP7 in T47D cells stably expressing shLARP7s. (**B**) qRT-PCR analysis of 7SK and LARP7 RNA levels in T47D cells stably expressing scramble control or shLARP7s. (**C–F**) Silencing LARP7 in T47D cells results in increased cell proliferation (**C**), anchorage-independent growth (**D**), cell migration (**E**), and invasion (**F**). (**G**) Representative images of mammospheres formed by control or shLARP7 cells. The numbers of primary and secondary mammospheres were

*Figure 3. Continued on next page*

*Figure 3. Continued*

quantified and shown in the graph below. Scale bars represent 100 μm. (**H**) qRT-PCR analysis of various epithelial and mesenchymal markers, MMPs, and stem cell markers in control and two shLARP7 pools. DSP, desmoplakin; KRT19, keratin 19. PCR values were normalized to that of β-actin. (**I**) Representative images of the H&E-stained lung sections from mice injected with control or shLARP7-2 cells. Scale bar, 100 μm. Quantification of the number of metastatic nodules is shown in the bar graph below. n = 6 in each group. Data in this figure are presented as the mean ± SD, and the p values (*p<0.05) were determined by the Student's *t* test.

The following figure supplement is available for figure 3:

**Figure supplement 1**. Silencing LARP7 accelerates malignant progression of BT474 breast cancer cells.

Ser2 in MCF10A and T47D cells at 0.3 μM (*Figure 4B*). Treatment of LARP7 KD cells with this concentration of flavopiridol fully blocked the increase in cell motility and migration (*Figure 4C,D*). In addition to the pharmacological inhibition of P-TEFb, silencing CDK9 expression by two different siRNAs also abolished the LARP7 KD-induced increase in cell migration (*Figure 4E*). Taken together, these data are consistent with the model that the elevated P-TEFb activity in the LARP7 KD cells is responsible for the observed increase in EMT and invasion.

## LARP7 KD increases expression of key EMT regulators through P-TEFb-dependent transcriptional activation

Because P-TEFb is a transcription elongation factor that most likely affects breast cancer progression at the level of transcription, we decided to examine the expression of a panel of EMT regulators in the two T47D LARP7 KD cell lines (shLARP7-1 and shLARP7-2). Consistent with the enhanced EMT phenotypes, loss of LARP7 markedly increased the expression of many key EMT and metastasis genes (*Zeisberg and Neilson, 2009*), including Slug, FOXC2, ZEB2, Twist1, ZEB1, Snail, Twist2, and SOX10 (*Figure 5A*). Notably, treatment of these LARP7 KD cells with flavopiridol or introduction of siCDK9s strongly inhibited the expression of four of those genes: Slug, FOXC2, ZEB2, and Twist1 (*Figure 5B,C*), suggesting that P-TEFb may directly affect their expression.

To determine whether these four genes are direct targets of P-TEFb, we performed the chromatin immunoprecipitation (ChIP) assay to measure the occupancy of CDK9 and pSer2 CTD at these gene loci at three different positions: the promoter-proximal/transcription start site (TSS), the interior of the open reading frame, and the 3′-untranslated region (UTR). As shown in *Figure 5D*, an enrichment of both CDK9 and pSer2 CTD was detected at all three positions in Slug, FOXC2, ZEB2, and Twist1 genes upon LARP7 depletion, but not in other non-responsive genes such as β-actin and ubiquitin C or EMT genes that are not direct targets of P-TEFb such as Snail and Twist2. These data suggest that LARP7 KD induces EMT by increasing P-TEFb-dependent expression of key EMT transcription factors.

It is worth noting that although some of the EMT genes such as Snail and Twist2 are not directly bound by CDK9, their expression may still be subjected to regulation by P-TEFb due to the fact that their transcription can be activated by Slug and Twist1, respectively (*Smit et al., 2009*; *Wels et al., 2011*; *Guo et al., 2012*), and therefore indirectly activated by P-TEFb.

## LARP7 KD shifts P-TEFb equilibrium to the transcriptionally active SEC

Since LARP7 depletion reduces 7SK snRNA levels, leading to disruption of the 7SK snRNP and P-TEFb activation, we next investigated whether the increased P-TEFb activity was due to redistribution of P-TEFb from the inhibitory 7SK snRNP to the transcriptionally active P-TEFb complexes. Toward this goal, CKD9 was immunoprecipitated from the nuclear extracts of T47D control and LARP7 KD cells, and its associated factors were analyzed by Western blotting. LARP7 KD decreased the association of P-TEFb with the 7SK snRNP components HEXIM1 and MePCE, while increased P-TEFb binding to the SEC components AFF4, ELL2, and AF9 (*Figure 6A*). The increased CDK9 levels in the SEC was most likely due to the enhanced transcription of the three SEC genes as revealed by qRT-PCR (*Figure 6B*), which in turn resulted in elevated levels of these proteins in the nuclear extracts (*Figure 6A*). Thus, LARP7 KD releases P-TEFb from the 7SK snRNP and redistributes it to the active SEC.

To determine whether the elevated SEC formation in LARP7 KD cells could increase SEC's association with key P-TEFb-targeted EMT genes, we performed the ChIP assay to examine the binding of the signature SEC subunit ELL2 to the genes encoding EMT transcription factors. As shown in *Figure 6C*, ELL2 was enriched on the four P-TEFb-responsive EMT genes Slug, FOXC2, ZEB2, and Twist1 upon

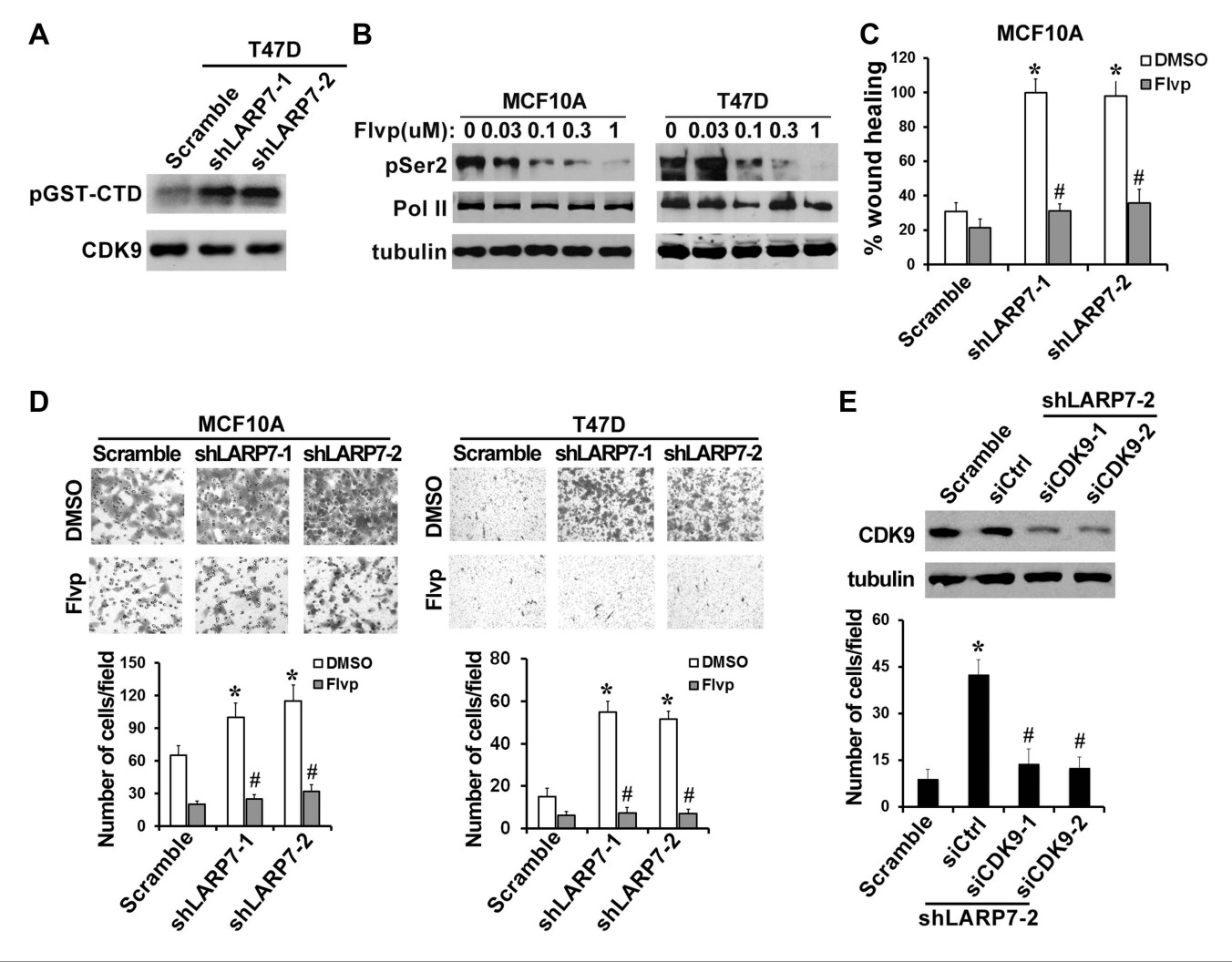

**Figure 4**. P-TEFb is required for the increased cell migration and invasion induced by LARP7 KD. (**A**) An in vitro kinase assay using GST-CTD of Pol II as an exogenous substrate shows that silencing LARP7 results in activation of P-TEFb kinase. (**B**) Flavopiridol (Flvp) inhibits Ser2 phosphorylation of Pol II CTD in MCF10A and T47D cells. Cells were treated with varying concentrations of flavopiridol for 7 hr before lysis, and pSer2 and total Pol II levels were analyzed by Western blotting. Tubulin was used as a loading control. (**C–D**) Treatment of cells with 0.3 μM flavopiridol reverses the accelerated cell motility of shLARP7 cells in the wound healing assay (**C**) and cell migration in the Transwell assay (**D**). Data are presented as the mean ± SD. p values were determined by the Student's t test. *p<0.05; comparison was between shLARP7 and scramble groups under DMSO treatment; #p<0.05; comparison was between Flvp- and DMSO-treated cells within the same cell lines. (**E**) Silencing CDK9 by siRNA reverses shLARP7-induced increase in cell migration. Two siCDK9s (siCDK9-1 and siCDK9-2) were transfected into T47D shLARP7-2 cells, and cell migration was assessed by a Transwell assay and quantified in the graph below. The efficiency of CDK9 knockdown was examined by Western blotting (upper panel). p values were determined by the Student's t test. *p<0.05, compared between shLARP7-2 siCtrl and scramble; #p<0.05, compared between shCDK9s and siCtrl.

LARP7 KD, but not on non-responsive β-actin and ubiquitin C genes or Snail and Twist2 that are not direct P-TEFb targets, strongly implicating the involvement of the SEC in supporting the LARP7 KD-induced EMT. Taken together, these data indicate that silencing LARP7 in noninvasive breast cancer cells shifts the P-TEFb equilibrium from the inhibitory 7SK snRNP to the active SEC, leading to increased P-TEFb activity and expression of EMT-related transcription factors.

## P-TEFb activity is essential for the survival and migration of malignant breast cancer cells

If LARP7 downregulation accelerates malignant progression by increasing expression of EMT genes through elevated P-TEFb activity, we should expect P-TEFb inhibition or re-introduction of LARP7

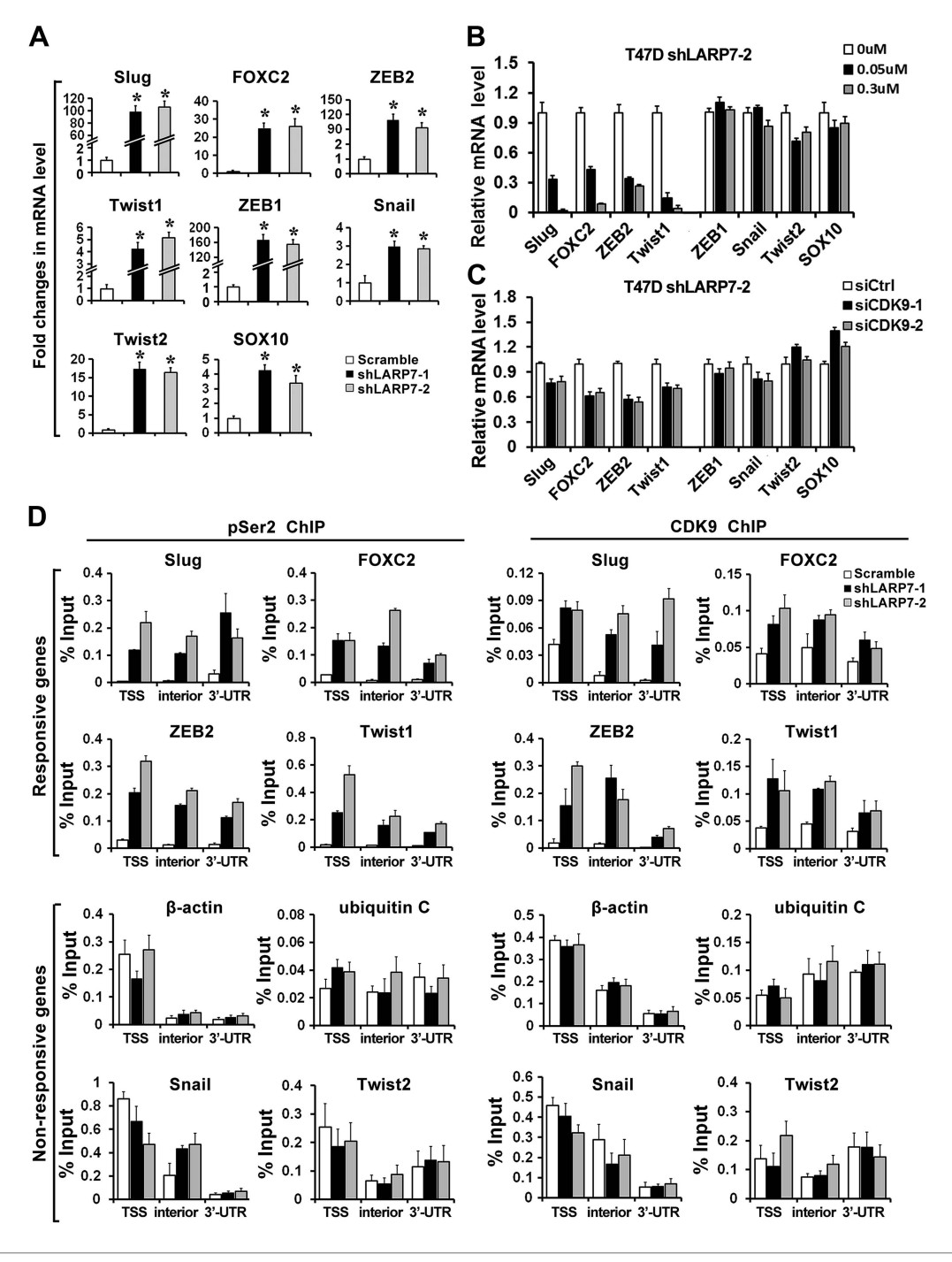

**Figure 5**. Loss of LARP7 enhances transcription of key EMT genes. (**A**) LARP7 knockdown significantly increases the expression of key EMT transcriptional factors in T47D breast cancer cells as measured by qRT-PCR. The p values (*p<0.05) were determined by the Student's *t* test. (**B**) Cells were treated with flavopiridol 7 hr at the indicated concentrations. The expression of EMT genes was assessed by qRT-PCR. (**C**) Cells were transfected with siCtrl, siCDK9-1 or siCDK9-2. After 48 hr, the expression of EMT genes was assessed by qRT-PCR. (**D**) Control or two shLARP7 T47D cells were subjected to ChIP analysis to determine the levels of CDK9 and phospho-Ser2 (pSer2) Pol II occupancy at various locations of EMT-related transcription factors and non-responsive genes. qRT-PCR was performed using primers specific to the transcription start site (TSS), interior and 3'-UTR of each gene, and the signals were normalized to that of input. Data are presented as the mean ± SD from three independent measurements.

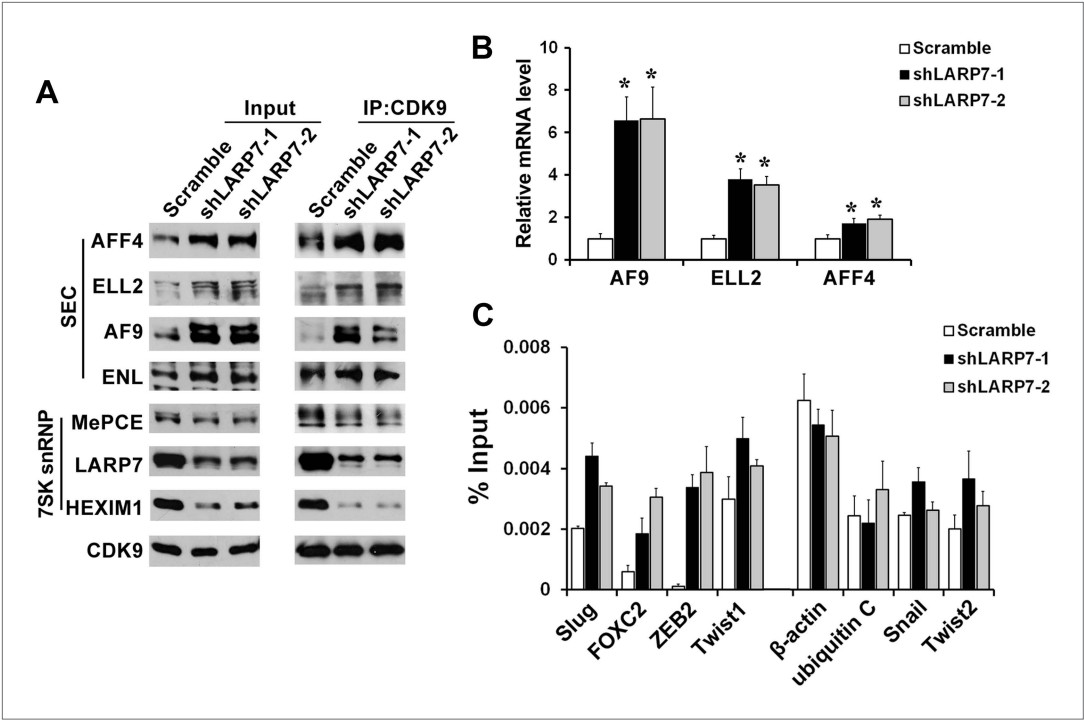

**Figure 6**. Silencing LARP7 redistributes P-TEFb from the 7SK snRNP to SEC. (**A**) Nuclear extracts were prepared from the control or shLARP7 cells and subjected to immunoprecipitation (IP) with anti-CDK9 antibodies. The 7SK snRNP and SEC formation was examined by Western blotting. (**B**) qRT-PCR analysis of AF9, ELL2, and AFF4 expression in T47D cells with or without LARP7 KD. The p values (*p<0.05) were determined by the Student's *t* test. (**C**) ChIP assay was performed to determine the binding of ELL2 to the EMT-related genes and non-responsive genes. Primers at the TSS of each gene were used. Data are presented as the mean ± SD from three independent measurements.

protein to block EMT and transformation. To test this hypothesis, we first treated the highly invasive and metastatic MDA-MB-231 breast cancer cells with flavopiridol and noted that the treatment significantly inhibited cell migration (*Figure 7A*). Furthermore, mRNA levels of Slug, a major EMT regulator, were repressed in a dose-dependent manner upon P-TEFb inhibition (*Figure 7B*). These data suggest that inhibition of P-TEFb can effectively block cell migration in metastatic breast cancer cells.

Next, to investigate whether re-introducing LARP7 in metastatic breast cancer cells could block or reverse the malignant phenotypes, we transfected MDA-MB-231 cells with cDNAs encoding either WT LARP7 or the LARP7 Δ2A mutant (*Figure 7C*). The Δ2A mutant was originally found in microsatellite instable gastric cancer and contains a deletion of two adenosines from a microsatellite repeat of 8 A's (nucleotides 1206–1213) in the LARP7 C-terminal region, leading to a frameshift deletion of the C-terminal region of LARP7. As a result, the Δ2A mutant cannot bind to 7SK snRNP and fails to suppress P-TEFb activity (*He et al., 2008*). The transfected cells were selected with puromycin and subjected to a colony formation assay. Interestingly, the LARP7-overexpressing cells formed significantly fewer colonies than those harboring the control vector (*Figure 7D*), suggest that re-expression of LARP7 impairs survival of these metastatic breast cancer cells. In contrast, cells expressing the Δ2A mutant formed as many colonies as the vector control cells, suggesting that the ability of LARP7 to assemble 7SK snRNP and suppress P-TEFb is necessary for its tumor suppressor potential. Taken together, these data indicate that LARP7 likely suppresses the survival and progression of malignant breast cancer cells through 7SK snRNP-dependent inhibition of P-TEFb.

## Discussion

In human cells, the 7SK snRNP represents the principle cellular reservoir of uncommitted P-TEFb, and its integrity is maintained through LARP7's direct interaction with and stabilization of the 7SK snRNA (*He et al., 2008*). Here, we show that in noninvasive human breast cancer cells, disruption of this

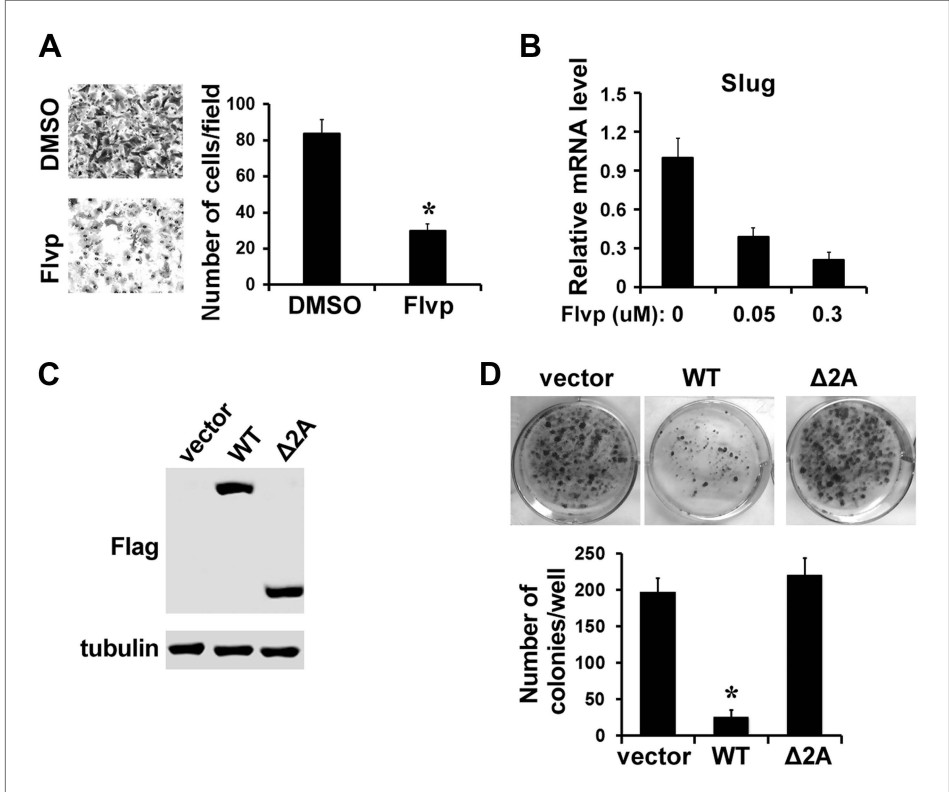

**Figure 7**. Inhibition of P-TEFb impairs EMT and survival of metastatic MDA-MB-231 cells. (**A**) Treatment of MDA-MB-231 cells with 0.3 µM flavopiridol (Flvp) significantly impairs cell migration as shown by the Transwell assay. The p value (*p<0.05) was determined by the Student's *t* test. (**B**) qRT-PCR analysis of Slug expression in MDA-MB-231 cells that have been treated with flavopiridol for 7 hr at the indicated concentrations. (**C** and **D**) Clonogenic growth assay. MDA-MB-231 cells were transfected with control vector, Flag-tagged WT LARP7, or Δ2A mutant. The expression of LARP7 and Δ2A was confirmed by Western blotting with anti-Flag. (**C**). (**D**) Representative culture areas are shown in the top panel. The number of colonies was quantified and shown in the graph below. The p value (*p<0.05) was determined by the Student's *t* test.

complex by knocking down LARP7 releases P-TEFb, redistributing it to the transcriptionally active SEC complex. This activation of P-TEFb promotes breast cancer EMT, invasion, cancer stem cell expansion, and metastasis by directly activating the expression of genes involved in EMT and metastasis. Thus, the sequestration of P-TEFb by 7SK snRNP is an effective anti-cancer mechanism, and the key 7SK snRNP component LARP7 is a potential tumor suppressor that specifically blocks breast cancer progression and metastasis. Our data highlight the importance of the transcription elongation machinery in regulating breast cancer EMT and metastasis and suggest a new therapeutic option to combat metastatic breast cancer through blocking P-TEFb activation. Currently, the CDK9 inhibitor flavopiridol is being evaluated in several phase I and II clinical trials for its anti-cancer effects either as a single agent or in combination with other drugs in treatment of esophageal cancer, B-cell chronic lymphocytic leukemia, endometrial carcinoma, recurrent/metastatic squamous cell carcinoma and most relevantly, previously treated locally advanced or metastatic breast cancer (http://www.cancernetwork.com/review-article/current-clinical-trials-flavopiridol/page/0/2). Thus, our studies are particular relevant and provide the mechanistic basis for targeting P-TEFb in patients with metastatic breast cancer.

LARP7 is a member of the LARP family that contains four La domain-containing RNA binding proteins (LARP1, 4, 6, and 7) with distinct RNA target preferences (*Bayfield et al., 2010*). Among the four members, LARP7 is the only one that binds to 7SK snRNA and is involved in transcription elongation controlled by Pol II. The role of the LARP family proteins in human cancer is so far poorly known, and our study is the first demonstration that LARP7 functions as a potential suppressor of human

breast cancer metastasis. This function of LARP7 is additionally supported by the observation that the *Drosophila* homolog of LARP7, multisex-combs (MXC), also acts as a tumor suppressor to inhibit cell growth (*Remillieux-Leschelle et al., 2002*). Moreover, the C-terminal region of LARP7, which is essential for the sequestration of P-TEFb in 7SK snRNP, is required for LARP7's suppression of breast cancer survival and is frequently deleted in human gastric cancer (*Mori et al., 2002*), suggesting that LARP7 may be a potential tumor suppressor in a broad range of human carcinomas. Finally, an inactivating mutation of LARP7 has been linked to a novel form of familial Primordial Dwarfism characterized by facial dysmorphism and intellectual disability (*Alazami et al., 2012*). In these patients, the 7SK snRNA is depleted due to the defective LARP7, indicating that LARP7 and 7SK snRNA, through their ability to control P-TEFb activity, also regulate embryonic development. Although it is formally possible that the LARP7 KD may also affect other yet-to-be-identified pathways in a P-TEFb-independent manner, our observation that both flavopiridol and siCDK9s efficiently rescued the increased cell migration of LARP7 KD cells lends a strong support to the notion that the observed effects of LARP7 KD on EMT and invasion are indeed mediated by P-TEFb activation.

A key observation of our study is that among the many EMT genes whose expression is altered by P-TEFb activation, the transcription factors Slug, FOXC2, ZEB2, and Twist1 are direct targets of P-TEFb and bound by the P-TEFb-containing SEC. These four proteins are well known EMT transcription factors that function as decisive early drivers of EMT. While FOXC2 orchestrates the mesenchymal component of the EMT program (*Mani et al., 2007*), Slug, ZEB2, and Twist1 repress E-cadherin expression, a fundamental event in EMT, as well as the expression of proteins of tight junctions, gap junctions, and desmosomes, contributing to the de-differentiated state of tumor cells and facilitating EMT (*Peinado et al., 2007*; *Sanchez-Tillo et al., 2012*). In addition, these proteins are also closely linked to breast cancer stem cell properties and implicated in resistance to drug and radio-therapies (*Voulgari and Pintzas, 2009*; *Singh and Settleman, 2010*; *Hollier et al., 2013*). Furthermore, these EMT early drivers also activate the expression of other EMT transcription factors. For example, Twist1 can induce Snail (*Smit et al., 2009*), and Slug can upregulate Twist2 and ZEB1 expression (*Wels et al., 2011*; *Guo et al., 2012*). Thus, by directly activating the expression of these EMT early drivers, P-TEFb is capable of orchestrating the EMT program to drive malignant progression. Since P-TEFb and its associated factors are expressed in most cell types, it is possible that the ability of P-TEFb to promote tumor progression may not be restricted to breast cancer but also true for other malignant tumors. Indeed, the active P-TEFb-Brd4 complex has been linked to AML and CML (*Dawson et al., 2011*; *Mertz et al., 2011*; *Zuber et al., 2011*; *Herrmann et al., 2012*; *Winter et al., 2012*), while multiple SEC components are direct targets of MLL-fusion proteins that are linked to various forms of leukemia (*Lin et al., 2010*; *Yokoyama et al., 2010*).

We have found that in the noninvasive breast cancer cell lines, P-TEFb released from the 7SK snRNP is mainly redistributed to the SEC. The expression levels of two SEC components AF9 and ELL2 are also markedly upregulated, allowing more SEC formation. Since ELL2 is another key catalytic component of SEC besides P-TEFb, its increase is likely to further enhance transcriptional elongation of target genes. We found that the Brd4-P-TEFb complex was also increased moderately upon LARP7 KD (data not shown), suggesting the possibility that Brd4 may also play a role in facilitating the P-TEFb-dependent breast cancer progression.

Given that P-TEFb is mostly known as a general transcription factor, our observations raise two interesting questions: Why are EMT transcription factors particularly sensitive to P-TEFb activation in these cells, and how is the SEC involved in this process? Our data that P-TEFb and SEC are associated with the genes encoding decisive EMT regulators but not non-responsive genes suggest that the direct target EMT genes may contain sequences that display high affinity binding to the SEC-P-TEFb complex. For example, these genes may contain the recently discovered Super-Enhancers that render them highly sensitive to activation by the Mediator and the SEC (*Hnisz et al., 2013*; *Loven et al., 2013*; *Whyte et al., 2013*). Interestingly, previous proteomic and biochemical analyses have identified the Mediator as a SEC binding partner through interaction between the Mediator subunit MED26 and the SEC components ELL-associated factor 1 (EAF1) and EAF2 (*Takahashi et al., 2011*), suggesting that the Mediator may recruit the SEC to target genes and the two complexes may act together to activate genes containing the Super-Enhancers. The SEC has also been linked to the elongating Pol II on chromatin template through interaction between the SEC components ENL/AF9 and the Pol II-associated factor 1 (PAF1) (*He et al., 2011*). Finally, SEC may be recruited to its target promoters through tissue-specific and sequence-specific DNA binding partners, such as the HIV Tat protein in

HIV-infected cells, MLL-fusion proteins in leukemia cells (*He and Zhou, 2011*; *Smith et al., 2011*) or retinoic acid receptor in differentiating mouse embryonic stem cells (*Lin et al., 2011*). Besides the positive control exerted by P-TEFb and SEC, another possibility is that the EMT genes are strongly suppressed by negative transcription elongation factors DSIF and NELF, and thus are very sensitive to P-TEFb activation. Future studies will try to determine which of these mechanisms may be responsible for the heightened sensitivity of EMT transcription factors to the elongation control.

Taken together, our study has demonstrated that LARP7 functions as a potential tumor suppressor in human breast cancer by suppressing the activity of P-TEFb and inhibiting EMT, invasion, and metastasis. Our study also provides the first demonstration that the transcription elongation machinery and the network of P-TEFb complexes play critical roles in regulating tumor EMT, cancer stemness, and metastasis by directly controlling the expression of upstream EMT regulators. These findings will facilitate the development of new drugs that target CDK9 for anti-cancer therapy.

## Materials and methods

### Cell lines and antibodies

The MCF10A mammary epithelial cells were cultured in DMEM-F12 medium supplemented with 5% horse serum, 20 ng/ml EGF, 10 μg/ml insulin, 0.5 μg/ml hydrocortisone, 100 ng/ml of cholera toxin, and penicillin/streptomycin. The EpH4 murine mammary epithelial cells were cultured as previously described (*Xu et al., 2009*). The breast cancer cells BT474, T47D, ZR75B, and BT549 were cultured in RPMI1640 media containing 10% FBS. MCF7, MDA-MB-468, MDA-MB-231, and MDA-MB-435 cells were cultured in DMEM plus 10% FBS.

The anti-LARP7 and anti-MePCE antibodies were generated as previously described (*He et al., 2008*; *Xue et al., 2010*). Antibodies against CDK9, CyclinT1, HEXIM1, and Brd4 have been described earlier (*Yik et al., 2003*; *Yang et al., 2005*). Antibodies against AFF4 and Pol II CTD repeat YSPTSPS (phospho S2) were purchased from Abcam (Cambridge, UK). Antisera against Pol II (8WG16) were purchased from Santa Cruz Biotechnology (Dallas, TX). The anti-ELL2 (A302-505A-1) antibodies were purchased from Bethyl Laboratories, Inc (Montgomery, TX). Antibodies against tubulin and E-cadherin were from Calbiochem (San Diego, CA) and BD (Franklin Lakes, NY), respectively. Anti-Flag antibody was purchased from Sigma (St. Louis, MO).

### Clinical data set analysis

The Oncomine database (www.oncomine.org) was searched for the expression profiles of P-TEFb and its associated factors. Only the data sets examining mRNA expression in cancer tissue with matched normal tissue controls (cancer vs normal) were included in this study. The threshold search criteria were $p$ value<0.05, fold change >2, and gene rank percentile <10%. p values presented in this study were calculated using a two-sided Student's *t*-test. NKI295 gene expression data were downloaded from the Stanford Microarray Database (http://microarray-pubs.stanford.edu/wound_NKI/explore.html). Survival data, stratified by expression of LARP7, were analyzed by using SPSS 13.0 and tested for significance using the log-rank test.

### Breast cancer tissue array analysis

The breast cancer tissue array (BR1503b) was purchased from US Biomax, Inc. Immunohistochemistry was carried out using the Tyramide Signal Amplification Biotin System Kit (PerkinElmer) with anti-LARP7 antibodies (10 μg/ml), following the manufacturer's instructions. Images were captured using the Zeiss AxioImager M2 microscope. The intensity of LARP7 stain (the number of pixels/area) in each sample was quantified by analyzing at least three stained areas using the ImageJ plus software. Statistical analysis was performed using the SPSS 13.0 program and determined by one-way ANOVA.

### Transfection and infection

The small hairpin RNA (shRNA) vector targeting human LARP7 was introduced into breast cancer cells by retroviral infection as described previously (*Zhu et al., 2007*). Briefly, shLARP7 in pSUPER-retro-puro vector was transfected together with retroviral packaging vectors into 293T cells, and viral supernatant was used to infect the breast cancer cells. Pools of infected cells were selected in the presence of puromycin and analyzed by a variety of assays. The shLARP7 sequences are: shLARP7-1, 5′-AATCACAGCTGGATTGAAA-3′; shLARP7-2, 5′-AAGTTAATCACCAAAGCTG-3′. A scrambled sequence having similar base compositions to the shLARP7s was used as a negative control. Lentivirus production

and infection were conducted as previously described (*Moffat et al., 2006*). The MCF10A shLARP7 stable cell lines and the rescue pool expressing the shRNA-resistant WT LARP7 have been reported previously (*He et al., 2008*). siRNAs targeting CDK9 were introduced into cell by transfection. Sequences of the two siCDK9 constructs used in the study are: siCDK9-1: 5′-GGGAGAUCAAGAUCCUUCATT-3′; siCDK9-2: 5′-GGUGAUGCAGAUGCUGCUUTT-3′.

## Quantitative RT-PCR (qRT-PCR)

qRT-PCR was performed with ABI 7300 (Applied Biosystem) and DyNAmo HS SYBR Green qPCR kit (Fisher Scientific) as per manufacturer's instruction. The gene-specific primers (*Supplementary file 1A*) were used at a final concentration of 0.2 μM. All PCR reactions were performed in triplicates.

## Preparation of nuclear extracts (NEs) and immunoprecipitation

NEs were prepared from various cell lines using a standard protocol (*Dignam et al., 1983*). Immunoprecipitation and Western blotting were performed as previously described (*He et al., 2008*). Briefly, NEs were incubated with specific antibodies at 4°C overnight and then with protein A beads (Invitrogen) for 2 hr. After washing with buffer D (20 mM HEPES-KOH [pH7.9], 15% glycerol, 0.2 mM EDTA, 0.2% NP-40, 1 mM dithiothreitol, and 1 mM phenylmethylsulfonyl fluride) containing 0.3 M KCl (D0.3M), the isolated proteins were eluted with 0.1 M glycine (pH 2.0) and analyzed by Western blotting with indicated antibodies.

## Chromatin immunoprecipitation (ChIP) assay

The ChIP assay was performed according to the manufacturer's instructions (EZ-ChIP: Catalog # 17-371; Millipore (Billerica, MA). Briefly, 2 μg antibodies per reaction were incubated with chromatin fractions in an immunoprecipitation assay, and the isolated chromatin fraction was purified and subjected to qRT-PCR using primers listed in *Supplementary file 1B*.

## In vitro kinase assay

The in vitro kinase assay was performed as described (*Yang et al., 2005*) with minor modifications. Briefly, CDK9 were isolated by immunoprecipitation from NEs of breast cancer cells, washed with buffer D0.15M for three times, and then subjected to the kinase assay using 4 μg GST-CTD as an exogenous substrate in the presence of 5uCi γ-[P32]-ATP. The phosphorylated GST-CTD was analyzed by SDS-PAGE and visualized by autoradiography.

## Wound healing assay

Cells were cultured in six-well plates to full confluence. A plastic tip was used to generate a wound across the cell monolayer. The wound closure was measured after 16 hr.

## Cell migration and invasion assays

Migration assays were performed in Transwell chambers (Corning). $1 \times 10^5$ cells in medium containing 1% FBS were seeded onto membranes in top chambers and allowed to migrate towards the bottom chambers filled with medium containing 10% FBS. Cells that remained in the upper chambers were removed with a cotton ball. Migrated cells were stained with crystal violet and photographed. Invasion assays were conducted using the BD Matrigel Invasion Chambers as per the manufacturer's protocol.

## Immunofluorescence staining

Cells were fixed in 4% paraformaldehyde for 20 min and permeabilized in 0.1% Triton X-100 for 5 min. Rhodamine-conjugated phalloidin (Molecular Probes: Eugene, OR) was employed to stain actin stress fiber (30 min at room temperature). E-cadherin was detected by staining with anti-E-cadherin antibodies (1 μg/ml) overnight at 4°C followed by incubating with Alexa 488-conjugated anti-mouse antibody (Molecular Probes) for 1 hr at room temperature. Cell nuclei were stained with Hoechst for 3 min.

## Soft agar assay

Growth medium (0.4 ml) containing 0.66% Bacto Agar (BD) was added to a 24-well plate in triplicates and allowed to harden. 2000 breast cancer cells were suspended in 0.2 ml medium containing 0.375% agar and overlaid on the hardened bottom layer. Fresh medium (0.2 ml) containing

0.375% agar was added to each well once a week for 4 weeks. The colonies were visualized by staining with 0.5 mg/ml 3-(4,5-dimethylthiazol-2-yl)-2,5-diphenyl tetrazolium bromide (MTT) (Sigma) for 4 hr at 37°C.

### Two-round mammosphere assay

Two-round mammosphere assay was performed as previously described (*Dontu et al., 2003*). Briefly, cells were trypsinized, counted, and plated on ultra-low attachment plates (Corning) at a density of 1000 cells/ml. Mammospheres were counted after 5 days. Cells were then dissociated, diluted, and re-seeded for a second round of mammosphere formation.

### In vivo lung metastasis assay

To evaluate the metastatic potential of shLARP7 cells, $2 \times 10^6$ cells in 150 µl serum-free medium were injected into the tail veins of 6-weeks-old female nude mice. After 12 weeks, mice were sacrificed, and quantitation of metastatic colonies was performed on representative hematoxylin and eosin (H&E)-stained sections of formalin-fixed and paraffin-embedded lungs.

### Colony formation assay

MDA-MB-231 cells were transfected with empty vectors, WT LARP7 or Δ2A cDNAs using Lipofectamine 2000 (Invitrogen). 48 hr after transfection, cells were seeded into six-well plates in triplicates and selected with puromycin (1.5 µg/ml) for 3 days. Surviving cells were continued to grow in medium without antibiotics. After 2 weeks, the colonies were stained with 0.1% crystal violet and quantified.

## Acknowledgements

We thank Nanhai He for gifts of LARP7 KD plasmids, Dapeng Wang for helpful assistance in analyzing tissue array data, and Qingwei Zhu and Jiangsha Zhao for technical assistance and discussion. This work was supported by Public Health Service grants R01AI095057 and R01AI041757 from the National Institutes of Health to QZ and R01DK090347 to KL.

## Additional information

### Funding

| Funder | Grant reference number | Author |
| --- | --- | --- |
| U.S. Public Health Service | R01AI095057 | Qiang Zhou |
| National Institutes of Health | R01AI041757 | Qiang Zhou |
| National Institutes of Health | R01DK090347 | Kunxin Luo |

The funders had no role in study design, data collection and interpretation, or the decision to submit the work for publication.

### Author contributions

XJ, Acquisition of data, Analysis and interpretation of data, Drafting or revising the article; HL, Acquisition of data, Analysis and interpretation of data; QZ, KL, Conception and design, Analysis and interpretation of data, Drafting or revising the article

### Ethics

Animal experimentation: All animals were handled according to the approved animal protocol (#R239) by the Institutional Animal Care and Use Committee (ACUC) of the University of California, Berkeley. All animal experiments were performed following the Guide for the Care and Use of Laboratory Animals of the National Institutes of Health.

## Additional files

### Supplementary file

• Supplementary file 1. List of qRT-PCR primers.

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
