## [Decision Letter]

Thank you for sending your work entitled “LARP7 suppresses P-TEFb activity to inhibit breast cancer progression and invasion” for consideration at *eLife*. Your article has been evaluated by Tony Hunter (Senior editor) and 3 reviewers, one of whom is a member of our Board of Reviewing Editors, and we invite you to submit a revised manuscript.

The Reviewing editor and the other reviewers discussed their comments before we reached this decision, and the Reviewing editor has assembled the following comments to help you prepare a revised submission.

The Reviewers agreed that this is a very interesting paper that links downregulation of the protein LARP7 to increased cell invasion and metastasis in breast cancer cells. The pioneering observation is that LARP7 is downregulated at the RNA and protein level in breast carcinomas relative to normal breast tissue, which correlates with poor prognosis. From there, the authors demonstrate that depletion of LARP7 in non-invasive breast cancer cells promotes an EMT-like phenotype, as shown by various cellular assays and molecular markers. Mechanistically, the authors show that LARP7 depletion leads to a redistribution of the transcription elongation factor P-TEFb from inactive to active nuclear complexes, concurrently with increased P-TEFb recruitment and RNAPII elongation at genes involved in EMT. The Reviewers would like to encourage submission of a revised manuscript addressing the major points below:

1) One major criticism is that the authors do not explain how increased P-TEFb activity in this system leads to selective activation of genes important for the EMT program and breast cancer progression. What makes free P-TEFb in this system selective for the genes controlling the expression of upstream EMT regulators over other RNA polymerase II protein-coding genes? Although the redistribution of free P-TEFb to the transcriptionally active SEC is well explained, it is unclear what dictates the specificity of P-TEFb recruitment to the genes involved in the EMT program. The molecular basis for the selectivity of this regulatory mechanism has to be better addressed and explained. Toward this end, Figure 5 should include gene expression and ChIP data for non-responsive genes. Figure 6 should also show data for non-responsive genes. As it stands, it is unclear how specific the results of gene induction upon LARP7 knockdown are.

2) Another major concern was the lack of an *in vivo* assay demonstrating that mere manipulation of LARP7 expression can change the metastatic potential of breast cancer cells. The *in vitro* cellular phenotypes are striking and the impact the paper would rise significantly by including experiments in murine models showing that LARP7 overexpression can block the metastatic ability of MDB-231 and/or that LARP7 depletion with shRNAs increases the metastatic potential of the non-invasive lines (e.g. MCF10A, T47D). If a negative result is obtained, this should be reported as well.

3) A third major concern was the depiction throughout the manuscript of LARP7 as a tumor suppressor. The studies were all carried out in cells and cell lines, not in cancer models. Thus the conclusion that LARP7 is a tumor suppressor is premature, it would be more conventionally characterized from this type of data as a 'potential tumor suppressor', especially in the absence of data that LARP mutations give rise to specific cancer types in mice and promote tumor growth or metastasis in tumor xenografts. The identification of some LARP7 mutations in some gastric cancers does not show causality. On this topic, the existing clinical trials that have evaluated flavopiridol for efficacy in metastatic solid tumors should be mentioned.

4) The experiments in Figure 2 should be complemented with a rescue assay to test whether an shRNA-resistant LARP7 construct can rescue the phenotype and revert cell morphology, migration and the wound healing properties as in the parental cell line.

5) Experiments in Figure 7 are highly relevant but incomplete. To support the overall model, it would be important to see whether re-expression of a LARP7 mutant with impaired 7SK RNA-binding activity (or the inactivating mutation linked to a novel form of Primordial Dwarfism), also impairs survival of the metastatic breast cancer cell (like wild type LARP7). It is expected that this mutant will not assemble into the snRNP and thus may not have tumor suppressor potential.

6) The EMT genes are shown to be upregulated in cells lacking LARP7, and are shown by ChIP to bind Cdk9, however the authors need to show that the expression of these genes requires Cdk9. Because flavopiridol inactivates several Cdks, specific shRNAs should be used to make this point.

---

## [Author Response]

The reviewers agreed that this is a very interesting paper that links downregulation of the protein LARP7 to increased cell invasion and metastasis in breast cancer cells. The pioneering observation is that LARP7 is downregulated at the RNA and protein level in breast carcinomas relative to normal breast tissue, which correlates with poor prognosis. From there, the authors demonstrate that depletion of LARP7 in non-invasive breast cancer cells promotes an EMT-like phenotype, as shown by various cellular assays and molecular markers. Mechanistically, the authors show that LARP7 depletion leads to a redistribution of the transcription elongation factor P-TEFb from inactive to active nuclear complexes, concurrently with increased P-TEFb recruitment and RNAPII elongation at genes involved in EMT. The Reviewers would like to encourage submission of a revised manuscript addressing the major points below:

We would like to thank the reviewers for the careful review and constructive suggestions. We have taken these comments very seriously and have taken the time to perform extensive experiments in order to address the reviewers’ concerns thoroughly. We believe that the manuscript has been significantly improved with these new additions.

*1) One major criticism is that the authors do not explain how increased P-TEFb activity in this system leads to selective activation of genes important for the EMT program and breast cancer progression. What makes free P-TEFb in this system selective for the genes controlling the expression of upstream EMT regulators over other RNA polymerase II protein-coding genes? Although the redistribution of free P-TEFb to the transcriptionally active SEC is well explained, it is unclear what dictates the specificity of P-TEFb recruitment to the genes involved in the EMT program. The molecular basis for the selectivity of this regulatory mechanism has to be better addressed and explained. Toward this end,*
Figure 5
*should include gene expression and ChIP data for non-responsive genes.*
Figure 6
*should also show data for non-responsive genes. As it stands, it is unclear how specific the results of gene induction upon LARP7 knockdown are*.

We have performed ChIP assay on two non-responsive Pol II-dependent genes, β-actin and ubiquitin C, and two non-flavopiridol-responsive EMT genes, Twist 2 and Snail, with anti-CDK9, anti-pSer2 (Figure 5) and anti-ELL2 (SEC) (Figure 6). As shown in the new Figure 5 and Figure 6, these non-responsive genes do not show increased binding of P-TEFb, phosphorylated Pol II and ELL2. Thus, one mechanism underlying the selectivity of P-TEFb for the upstream EMT regulators over other Pol II-dependent genes is the more efficient or selective binding of the P-TEFb complex to the former. As for what factors determine the efficiency/selectivity of P-TEFb binding, it is one of the most actively researched questions in the field. Recruitment of P-TEFb to its target genes probably involves Brd4 via its bromodomain, or more likely in our case, the SEC complex via several possible mechanisms such as the Mediator bound to the newly discovered Super enhancers or the Pol II-associated PAF complex. Finally, certain genes that are strongly associated with negative elongation factors NELF and DSIF may become especially sensitive to P-TEFb activation. We have included a brief discussion of these possibilities in the Discussion.

*2) Another major concern was the lack of an in vivo assay demonstrating that mere manipulation of LARP7 expression can change the metastatic potential of breast cancer cells. The in vitro cellular phenotypes are striking and the impact the paper would rise significantly by including experiments in murine models showing that LARP7 overexpression can block the metastatic ability of MDB-231 and/or that LARP7 depletion with shRNAs increases the metastatic potential of the non-invasive lines (e.g. MCF10A, T47D). If a negative result is obtained, this should be reported as well*.

We have performed *in vivo* metastasis assay by injecting T47D control and shLARP7-2 cells in the tail veins of the nude mice. 12 weeks after the injection, histological analysis was carried out to detect the presence of metastasis in the lung. Consistent with the increased EMT and invasion *in vitro*, the shLARP7-2 cells formed more and larger metastatic colonies in the lung compared to the control cells. Thus, LARP7 depletion increases the metastatic potential *in vivo*. These data have been added to Figure 3.

*3) A third major concern was the depiction throughout the manuscript of LARP7 as a tumor suppressor. The studies were all carried out in cells and cell lines, not in cancer models. Thus the conclusion that LARP7 is a tumor suppressor is premature, it would be more conventionally characterized from this type of data as a 'potential tumor suppressor', especially in the absence of data that LARP mutations give rise to specific cancer types in mice and promote tumor growth or metastasis in tumor xenografts. The identification of some LARP7 mutations in some gastric cancers does not show causality. On this topic, the existing clinical trials that have evaluated flavopiridol for efficacy in metastatic solid tumors should be mentioned*.

We have revised the text regarding LARP7 as a “tumor suppressor” to a “potential tumor suppressor”. We have included the following comment on the ongoing clinical trials of flavopiridol in malignant human cancer in the Discussion section:

“Currently, the CDK9 inhibitor flavopiridol is being evaluated in several phase I and II clinical trials for its anti-cancer effects either as a single agent or in combination with other drugs in treatment of esophageal cancer, B-cell chronic lymphocytic leukemia, endometrial carcinoma, recurrent/metastatic squamous cell carcinoma and most relevantly, previously treated locally advanced or metastatic breast cancer.”

*4) The experiments in*
Figure 2
*should be complemented with a rescue assay to test whether an shRNA-resistant LARP7 construct can rescue the phenotype and revert cell morphology, migration and the wound healing properties as in the parental cell line*.

We have performed the rescue experiment by introducing an shRNA-resistant WT LARP7 stably into the MCF10A shLARP7-2 cells. When expressed at a level comparable to that in MCF10A parental cells, this shRNA-resistant LARP7 readily restored the loss of 7SK snRNA (Figure 2), reverted the morphology of the cells to the epithelial-like (Figure 2) and mitigated the enhanced cell migration and motility (Figure 2), confirming that the increased EMT and cell migration are due to the loss of LARP7. We have incorporated these results in the new Figure 2.

*5) Experiments in*
Figure 7
*are highly relevant but incomplete. To support the overall model, it would be important to see whether re-expression of a LARP7 mutant with impaired 7SK RNA-binding activity (or the inactivating mutation linked to a novel form of Primordial Dwarfism), also impairs survival of the metastatic breast cancer cell (like wild type LARP7). It is expected that this mutant will not assemble into the snRNP and thus may not have tumor suppressor potential*.

We have performed the experiment suggested by the reviewers by introducing a LARP7 mutant Δ2A that was originally found in microsatellite instable gastric cancer and contains a deletion of 2 adenosines from a microsatellite repeat of 8 A’s (nucleotides 1206–1213) in the LARP7 C-terminal region. This results in a frame shift deletion of the C-terminal region of LARP7. The Δ2A mutant cannot bind to 7SK snRNP and therefore fails to suppress P-TEFb activity ([12], Mol Cell). When introduced into the MDA-MB-231 cells, the Δ2A mutant did not induce apoptosis of these cells whereas WT LARP7 readily impaired cell survival, supporting the model that the ability of LARP7 to assemble 7SK snRNP and suppress P-TEFb is necessary for its tumor suppressor potential. We have included these data in the new Figure 7.

*6) The EMT genes are shown to be upregulated in cells lacking LARP7, and are shown by ChIP to bind Cdk9, however the authors need to show that the expression of these genes requires Cdk9. Because flavopiridol inactivates several Cdks, specific shRNAs should be used to make this point*.

We have employed two different siRNAs against CDK9 to reduce its expression in T47D shLARP7-2 cells and examined their impacts on the expression of EMT transcription factors by qRT-PCR. Consistent with the results obtained by flavopiridol treatment (Figure 6), reducing CDK9 expression impaired the expression of 4 genes, Slug, FOXC2, ZEB2 and Twist1, but had no effect on that of other 4 genes. The new results with siCDK9s have been added to the new Figure 6.